# RISK AVERSE VALUE EXPANSION FOR SAMPLE EFFICIENT AND ROBUST POLICY LEARNING

## ABSTRACT

Model-based Reinforcement Learning(RL) has shown great advantage in sample-efficiency, but suffers from poor asymptotic performance and high inference cost. A promising direction is to combine model-based reinforcement learning with model-free reinforcement learning, such as model-based value expansion(MVE). However, the previous methods do not take into account the stochastic character of the environment, thus still suffers from higher function approximation errors. As a result, they tend to fall behind the best model-free algorithms in some challenging scenarios. We propose a novel Hybrid-RL method, which is developed from MVE, namely the Risk Averse Value Expansion(RAVE). In the proposed method, we use an ensemble of probabilistic models for environment modeling to generate imaginative rollouts, based on which we further introduce the aversion of risks by seeking the lower confidence bound of the estimation. Experiments on different environments including MuJoCo and robo-school show that RAVE yields state-of-the-art performance. Also we found that it greatly prevented some catastrophic consequences such as falling down and thus reduced the variance of the rewards.

## 1 INTRODUCTION

In contrast to the tremendous progress made by model-free reinforcement learning algorithms in the domain of games(Mnih et al., 2015; Silver et al., 2017; Vinyals et al., 2019), poor sample efficiency has risen up as a great challenge to RL, especially when interacting with the real world. Toward this challenge, a promising direction is to integrate the dynamics model to enhance the sample efficiency of the learning process(Sutton, 1991; Calandra et al., 2016; Kalweit & Boedecker, 2017; Oh et al., 2017; Racanière et al., 2017). However, classic model-based reinforcement learning(MBRL) methods tend to lag behind the model-free methods(MFRL) asymptotically, especially in cases of noisy environments and long trajectories. The hybrid combination of MFRL and MBRL(Hybrid-RL for short) has attracted much attention due to this reason. A lot of efforts has been devoted to this field, including the dyna algorithm(Kurutach et al., 2018), model-based value expansion(Feinberg et al., 2018), I2A(Weber et al., 2017), etc.

The robustness of the learned policy is another concern in RL. For stochastic environments, policy can be vulnerable to tiny disturbance and occasionally drop into catastrophic consequences. In MFRL, off-policy RL(such as DQN, DDPG) typically suffers from such problems, which in the end leads to instability in the performance including sudden drop in the rewards. To solve such problem, risk sensitive MFRL not only maximize the expected return, but also try to reduce those catastrophic outcomes(Garcıa & Fernández, 2015; Dabney et al., 2018a; Pan et al., 2019). For MBRL and Hybrid-RL, without modeling the uncertainty in the environment(especially for continuous states and actions), it often leads to higher function approximation errors and poorer performances. It is proposed that complete modeling of uncertainty in transition can obviously improve the performance(Chua et al., 2018), however, reducing risks in MBRL and Hybrid-RL has not been sufficiently studied yet.

In order to achieve both sample efficiency and robustness at the same time, we propose a new Hybrid-RL method more capable of solving stochastic and risky environments. The proposed method, namely Risk Averse Value Expansion(RAVE), is an extension of the model-based value expansion(MVE)(Feinberg et al., 2018) and stochastic ensemble value expansion(STEVE)(Buckman

et al., 2018). We systematically analyse the approximation errors of different methods in stochastic environments. We borrow ideas from the uncertainty modeling( Chua et al. (2018)) and risk averse reinforcement learning. The probabilistic ensemble environment model is used, which captures not only the variance in estimation(also called epistemic uncertainty), but also stochastic transition nature of the environment(also called aleatoric uncertainty). Utilizing the ensemble of estimations, we further adopt a dynamic confidence lower bound of the target value function to make the policy more risk-sensitive. We compare RAVE with prior MFRL and Hybrid-RL baselines, showing that RAVE not only yields SOTA expected performance, but also facilitates the robustness of the policy.

## 2 RELATED WORKS

The **model-based value expansion**(MVE)(Feinberg et al., 2018) is a Hybrid-RL algorithm. Unlike typical MFRL such as DQN that uses only 1 step bootstrapping, MVE uses the imagination rollouts of length $H$ to predict the target value. The assistance of environment model can greatly improve the sample efficiency at the start, but the precision of long term inference becomes limited asymptotically. In order to properly balance the contribution of the value expansion of different horizons, stochastic ensemble value expansion(STEVE)(Buckman et al., 2018) adopts an interpolation of value expansion of different horizon. The accuracy of the expansion is estimated through the ensemble of environment models as well as value functions. Ensemble of environment models also models the uncertainty to some extent, however, ensemble of deterministic model captures mainly epistemic uncertainty instead of stochastic transitions(Chua et al., 2018).

The **uncertainty** or the function approximation error is typically divided into three classes(Geman et al., 1992): the *noise* exists in the objective environment, e.g., the stochastic transitions, which is also called aleatoric uncertainty(Chua et al., 2018). The *model bias* is the error produced by the limited expressive power of the approximating function, which is measured by the expectation of ground truth and the prediction of the model, in case that infinite training data is provided. The *variance* is the uncertainty brought by insufficient training data, which is also called epistemic uncertainty. Dabney et al. (2018b) discuss the epistemic and aleatoric uncertainty in their work and focus on the latter one to improve the distributional RL. Recent work suggests that ensemble of probabilistic model(PE) is considered as more thorough modeling of uncertainty(Chua et al., 2018), while simply aggregate deterministic model captures only variance or epistemic uncertainty. The stochastic transition is more related to the noise(or aleatoric uncertainty), and the epistemic uncertainty is usually of interest to many works in terms of exploitation&exploration(Pathak et al., 2017; Schmidhuber, 2010; Oudeyer & Kaplan, 2009). Other works adopt ensemble of deterministic value function for exploration(Osband et al., 2016; Buckman et al., 2018).

**Risks** in RL typically refer to the inherent uncertainty of the environment and the fact that policy may perform poorly in some cases(García & Fernández, 2015). Risk sensitive learning requires not only maximization of expected rewards, but also lower variances and risks in performance. Toward this object, some works adopt the variance of the return(Sato et al., 2001; Pan et al., 2019; Reddy et al., 2019), or the worst-case outcome(Heger, 1994; Gaskett, 2003) in either policy learning(Pan et al., 2019; Reddy et al., 2019), exploration(Smirnova et al., 2019), or distributional value estimates(Dabney et al., 2018a). An interesting issue in risk reduction is that reduction of risks is typically found to be conflicting with exploration and exploitation that try to maximize the reward in the long run. Authors in (Pan et al., 2019) introduce two adversarial agents(risk aversion and long-term reward seeking) that act in combination to solve this problem. Still, it remains quite tricky and empiristic to trade-off between risk-sensitive and risk-seeking(exploration) in RL. In this paper, we propose a dynamic confidence bound for this purpose.

A number of prior works have studied function approximation error that leads to overestimation and sub-optimal solution in MFRL. Double DQN(DDQN)(Van Hasselt et al., 2016) improves over DQN through disentangling the target value function and the target policy that pursues maximum value. In TD3(Fujimoto et al., 2018) the authors suggest that systematic overestimation of value function also exists in actor-critic MFRL. They use an ensemble of two value functions, with the minimum estimate being used as the target value. Selecting the lower value estimation is similar to using uncertainty or lower confidence bound which is adopted by the other risk sensitive methods(Pan et al., 2019), though they claimed different motivations.

## 3 PRELIMINARIES

### 3.1 ACTOR-CRITIC MODEL-FREE REINFORCEMENT LEARNING

The Markov Decision Processes(MDP) is used to describe the process of an agent interacting with the environment. The agent selects the action $a_t \in \mathcal{A}$ at each time step $t$. After executing the action, it receives a new observation $s_{t+1} \in \mathcal{S}$ and a feedback $r_t \in \mathbb{R}$ from the environment. As we focus mainly on the environments of continuous action, we denote the parametric deterministic *policy* that the agent uses to decide its action as $a_t = \mu_\theta(s_t)$. Typically we add Gaussian exploration noise on top of the deterministic policy, such that we have a stochastic behavioral policy $\pi_{\theta,\sigma}$ : $\mathcal{S} \times \mathcal{A} \to \mathbb{R}$. It is calculated as $\pi_{\theta,\sigma}(s_t, a_t) = p_\mathcal{N}(a_t | \mu_\theta(s_t), \sigma^2)$, where $p_\mathcal{N}(x|m, \sigma^2)$ represents the probability density at $x$ in a Gaussian distribution $\mathcal{N}(m, \sigma^2)$. As the interaction process continues, the agent generates a trajectory $\tau = (s_0, a_0, r_0, s_1, a_1, r_1, ...)$ following the policy $\pi_{\theta,\sigma}$. For finite horizon MDP, we use the indicator $d : \mathcal{S} \to \{0, 1\}$ to mark whether the episode is terminated. The objective of RL is to find the optimal policy $\pi^*$ to maximize the expected discounted sum of rewards along the trajectory. The value performing the action $a$ with policy $\pi$ at the state $s$ is defined by $Q^\pi(s, a) = \mathbb{E}_{s_0=s, a_0=a, \tau \sim \pi} \sum_{t=0}^\infty \gamma^t r_t$, where $0 < \gamma < 1$ is the discount factor. The value iteration in model-free RL tries to approximate the optimal value $Q^{\pi^*}$ with a parametric value function $\hat{Q}_\phi$ by minimizing the Temporal Difference(TD) error, where $\phi$ is the parameter to be optimized. The TD error between the estimates of Q-value and the corresponding target values is shown in equation. 1, where $\phi'$ is a delayed copy of the parameter $\phi$, and $a' \sim \pi_{\theta'}$, with $\theta'$ being a delayed copy of $\theta$(Lillicrap et al., 2015).

$$\mathcal{L}_\phi = \mathbb{E}_\tau \left[ \sum_t (\hat{Q}_{target}(r_t, s_{t+1}) - \hat{Q}_\phi(s_t, a_t))^2 \right] \tag{1}$$
$$\text{with } \hat{Q}_{target}(r_t, s_{t+1}) = r_t + \gamma \cdot (1 - d(s_{t+1})) \cdot \hat{Q}_{\phi'}(s_{t+1}, a'))$$

To optimize the deterministic policy function in a continuous action space, deep deterministic policy gradient(DDPG)(Lillicrap et al., 2015) maximizes the value function (or minimizes the negative value function) under the policy $\mu_\theta$ with respect to parameter $\theta$, shown in equation. 2.

$$\mathcal{L}_\theta = -\mathbb{E}_\tau [\sum_t \hat{Q}_{\phi'}(s_t, \mu_\theta(s_t))] \tag{2}$$

### 3.2 ENVIRONMENT MODELING

To model the environment in continuous space, an environment model is typically composed of three individual mapping functions: $\hat{f}_{r,\zeta_r} : \mathcal{S} \times \mathcal{A} \times \mathcal{S} \to \mathbb{R}$, $\hat{f}_{s,\zeta_s} : \mathcal{S} \times \mathcal{A} \to \mathcal{S}$, and $\hat{f}_{d,\zeta_d} : \mathcal{S} \to [0, 1]$, which are used to approximate the feedback, next state and probability of the terminal indicator respectively(Gu et al., 2016; Feinberg et al., 2018). Here $\zeta_r$, $\zeta_s$ and $\zeta_d$ are used to represent the parameters of the corresponding mapping functions. With the environment model, starting from $s_t, a_t$, we can predict the next state and reward by

$$\hat{s}_{t+1} = \hat{f}_{s,\zeta_s}(s_t, a_t), \hat{r}_t = \hat{f}_{r,\zeta_r}(s_t, a_t, \hat{s}_{t+1}), \hat{d}_{t+1} = \hat{f}_{d,\zeta_d}(\hat{s}_{t+1}), \tag{3}$$

and this process might go on to generate a complete *imagined trajectory* of $[s_t, a_t, \hat{r}_t, \hat{s}_{t+1}, ...]$.

The neural network is commonly used as an environment model due to its powerful express ability. To optimize the parameter $\zeta$ we need to minimize the mean square error(or the cross entropy) of the prediction and the ground truth, given the trajectories $\tau$ under the behavioral policy.

### 3.3 UNCERTAINTY AWARE PREDICTION

The deterministic model approximates the *expectation* only. As we mentioned in the previous sections, overestimation is attributed to the error in function approximation. Following Chua et al. (2018), we briefly review different uncertainty modeling techniques.

**Probabilistic models** output a distribution (e.g., mean and variance of a Gaussian distribution) instead of an expectation. We take the reward component of the environment model as an example,

the probabilistic model is written as $r \sim \mathcal{N}(\hat{f}_{r,\zeta_r}, \hat{\sigma}^2_{r,\zeta_r})$, and the loss function is the negative log likelihood(equation. 4).

$$\mathcal{L}_{\zeta_r} = -\mathbb{E}_\tau \left[ \log p_\mathcal{N}(r_t | \hat{f}_{r,\zeta_r}(s_t, a_t, s_{t+1}), \hat{\sigma}^2_{r,\zeta_r}(s_t, a_t, s_{t+1})) \right] \tag{4}$$

**Ensemble of deterministic**(DE) models maintains an ensemble of parameters, which is typically trained with different training samples and different initial parameters. E.g, given the ensemble of parameters $\zeta_{r,1}, \zeta_{r,2}, ..., \zeta_{r,N}$, the expectation and the variance of the prediction is acquired from equation. 6

$$\hat{\mathbb{E}}_{\zeta_r} \left[ \hat{f}_{r,\zeta_r} \right] = \frac{1}{N} \sum_i \hat{f}_{r,\zeta_{r,i}}, \quad \hat{\mathbb{V}}_{\zeta_r} \left[ \hat{f}_{r,\zeta_r} \right] = \frac{1}{N} \sum_i (\hat{f}_{r,\zeta_{r,i}} - \hat{\mathbb{E}}_{\zeta_r} \left[ \hat{f}_{r,\zeta_r} \right])^2 \tag{5}$$

We define the operator $\hat{\mathbb{E}}_x$ and $\hat{\mathbb{V}}_x$ as the average and the variance on the ensemble of $x$es respectively. As proposed by (Chua et al., 2018), the variance $\hat{\sigma}^2$ in equation. 4 mainly captures the *aleatoric* uncertainty, and the variance $\hat{\mathbb{V}}$ mainly captures the *epistemic* uncertainty.

**Ensemble of probabilistic models**(PE) keeps track of an collection of distributions $\{\mathcal{N}(\hat{f}_{r,\zeta_{r,i}}, \hat{\sigma}^2_{r,\zeta_{r,i}})\}, i \in [1, N]$, which can further give the estimation of both uncertainties. A sampling form PE goes as follow

$$\text{Sample } i \text{ uniformly from } [1, N], \text{ then sample } r \sim \mathcal{N}(\hat{f}_{r,\zeta_{r,i}}, \hat{\sigma}^2_{r,\zeta_{r,i}}) \tag{6}$$

### 3.4 MODEL-BASED VALUE EXPANSION

In MVE(Feinberg et al., 2018) the learned environment model $\hat{f}_{\zeta_r, \zeta_s, \zeta_d} = (\hat{f}_{s,\zeta_s}, \hat{f}_{r,\zeta_r}, \hat{f}_{d,\zeta_d})$ together with the policy $\mu_\theta$ are used to image a trajectory starting from state $s_t$ and action $a_t$, which is represented by $\hat{\tau}_{\zeta_r, \zeta_s, \zeta_d, \theta', H}(r_t, s_{t+1})$. It produces a trajectory up to horizon $H(H \geq 0)$. We can write $\hat{\tau}_{\zeta_r, \zeta_s, \zeta_d, \theta', H}(r_t, s_{t+1}) = (r_t, s_{t+1}, \hat{a}_{t+1}, \hat{r}_{t+1}, \hat{s}_{t+2}, ..., \hat{s}_{t+H+1}, \hat{a}_{t+H+1})$.

Then the target value $\hat{Q}_{target}$ in equation. 2 is replaced with estimated return $\hat{Q}^{\text{MVE}}_{\zeta_r, \zeta_s, \zeta_d, \theta', \phi', H}$ on the sampled trajectory $\hat{\tau}_{\zeta_r, \zeta_s, \zeta_d, \theta', H}(r_t, s_{t+1})$, which is expressed in equation 7.

$$\hat{Q}_{target}(r_t, s_{t+1}) \leftarrow \hat{Q}^{\text{MVE}}_{\zeta_r, \zeta_s, \zeta_d, \theta', \phi', H}(r_t, s_{t+1})$$
$$= r_t + \sum_{t'=t+1}^{t+H} \gamma^{t'-t} d_{t,t'} \hat{r}_{t'} + \gamma^{H+1} d_{t,t+H+1} \hat{Q}_{\phi'}(\hat{s}_{t+H+1}, \hat{a}_{t+H+1}) \tag{7}$$
$$\text{with } d_{t,t'} = (1 - d(s_{t+1})) \prod_{k=t+2}^{t'} (1 - \hat{f}_{d,\zeta_d}(\hat{s}_k))$$

### 3.5 STOCHASTIC ENSEMBLE VALUE EXPANSION

Selecting proper horizon $H$ for value expansion is important to achieve high sample efficiency and asymptotic accuracy at the same time. Though the increase of $H$ brings increasing prophecy, the asymptotic accuracy is sacrificed due to the increasing reliance on the environment model. In STEVE(Buckman et al., 2018), interpolating the value expansions $\hat{Q}^{\text{MVE}}_{\zeta_r, \zeta_s, \zeta_d, \theta', \phi', H}$ of different $H \in [0, H_{max}]$ is proposed. The weight for each horizon is decided by the inverse of its variance, which is calculated by estimating an ensemble of values switching the combination of parameters in environment model $\hat{f}$ and value function $\hat{Q}_{\phi'}$. Through our notation, STEVE can be written as equation. 8.

$$\hat{Q}^{\text{STEVE}}(r_t, s_{t+1}) = \frac{\sum_{H=0}^{H_{max}} \omega_H \hat{\mathbb{E}}_{\zeta_r, \zeta_s, \zeta_d, \phi'} \left[ \hat{Q}^{\text{MVE}}_{\zeta_r, \zeta_s, \zeta_d, \theta', \phi', H}(r_t, s_{t+1}) \right]}{\sum_{H=0}^{H_{max}} \omega_H}, \tag{8}$$
$$\text{with } \omega_H = \hat{\mathbb{V}}_{\zeta_r, \zeta_s, \zeta_d, \phi'} \left[ \hat{Q}^{\text{MVE}}_{\zeta_r, \zeta_s, \zeta_d, \theta', \phi', H}(r_t, s_{t+1}) \right]^{-1}$$

## 4 INVESTIGATION OF THE APPROXIMATION ERROR IN STOCHASTIC ENVIRONMENTS

To thoroughly investigate the impact of aleatoric uncertainty on hybri-RL methods, we construct a demonstrative toy environment(fig. 1(a)). The agent starts from $s_0 = 0$, chooses an action $a_t$ from $\mathcal{A} = [-1, +1]$ at each time step $t$. The transition of the environment is $s_{t+1} = s_t + \frac{a_t}{|a_t|} + k \cdot \mathcal{N}(0, 1)$. We compare two different environments:$k = 0$ and $k = 1$, where $k = 0$ represents the deterministic transition, and $k = 1$ represents the stochastic transition. The episode terminates at $(|s| > 5)$, where the agent acquires a final reward. The agent gets a constant penalty(-100) at each time step to encourage it to reach the terminal state as soon as possible. Note that the deterministic environment actually requires more steps in expectation to reach $|s > 5|$ compared with the stochastic environment, thus the value function at the starting point of $k = 1$ (Ground truth = 380+) tends to be lower than that of $k = 0$(Ground truth = 430+).

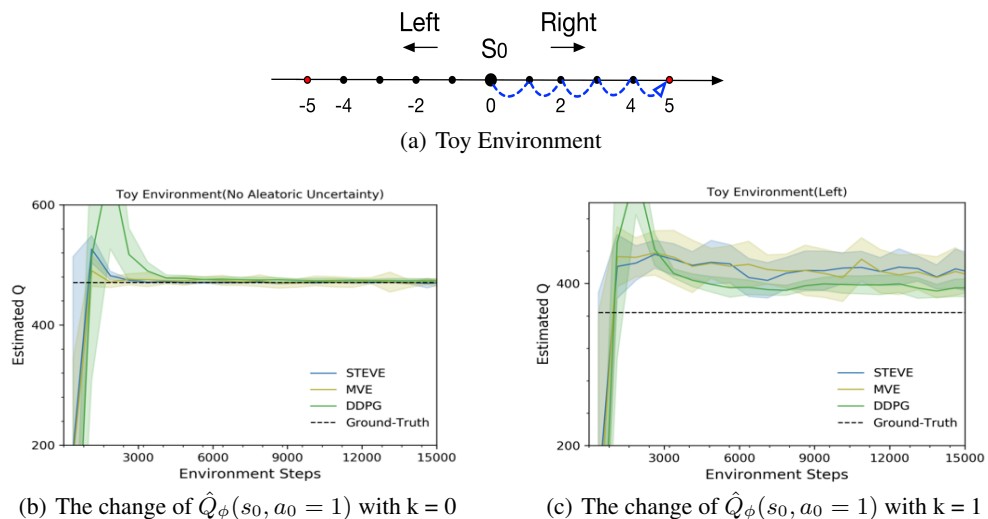

(a) Toy Environment

(b) The change of $\hat{Q}_\phi(s_0, a_0 = 1)$ with k = 0

(c) The change of $\hat{Q}_\phi(s_0, a_0 = 1)$ with k = 1

Figure 1: Curves of estimated Q values in the toy environment at the starting point over the environment steps.Each experiment is run four times.

We apply the learning methods including DDPG, MVE, STEVE to this environment, and plot the changes of estimate values at the starting point(see fig. 5).

The results show that, in the deterministic environment, the Q-values estimated by all methods converge to the ground-truth asymptotically in such a simple environment. However, after adding the noise, previous MFRL and Hybrid-RL methods show various level of overestimation. The authors of (Feinberg et al., 2018) have claimed that value expansion improves the quality of estimated values, but MVE and STEVE actually give even worse prediction than model-free DDPG in the stochastic environment. A potential explanation is that the overall overestimation comes from the unavoidable imprecision of the estimator(Thrun & Schwartz, 1993; Fujimoto et al., 2018), but Hybrid-RL also suffers from the approximation error of the dynamics model. When using a deterministic environment model, the predictive transition of both environments would be identical, because the deterministic dynamics model tends to approximate the expectation of next states(e.g, $\hat{f}_{s,\zeta_s}(s_t = 0, a_t > 0) = 1.0, \hat{f}_{s,\zeta_s}(s_t = 1.0, a_t > 0) = 2.0$). This would result in the same value estimation for $k = 0$ and $k = 1$ for both value expansion methods, but the ground truth of Q-values are different in these two environments. As a result, the deterministic environment introduces additional approximation error, leading to more severe overestimation.

## 5 METHODOLOGY

### 5.1 RISK AVERSE VALUE EXPANSION

We proposed mainly two improvements based on MVE and STEVE. Firstly, we apply an ensemble of probabilistic models (PE) to enable the environment model to capture the uncertainty over possible trajectories. Secondly, inspired by risk sensitive RL, we calculate the confidence lower bound of the target value(fig 1(c)).

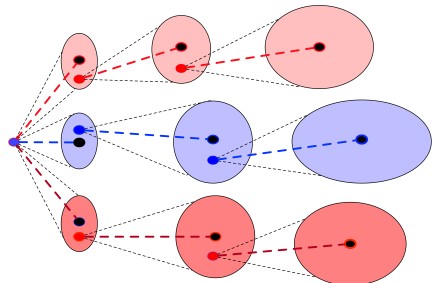

Figure 2: A casual illustration of three Distributional Value Expansion(DVE) trajectories.

Before introducing RAVE we start with the Distributional Value Expansion(DVE). Compared with MVE that uses a deterministic environment model and value function, DVE uses a probabilistic environment model, and we independently sample the reward and the next state using the probabilistic environment models(see equation 9 and fig. 2).

$$
\begin{aligned}
\tilde{s}_{t+2} &\sim \mathcal{N}(\hat{f}_{s,\zeta_s}(s_{t+1}, \tilde{a}_{t+1} = \mu_{\theta'}(s_{t+1})), \hat{\sigma}^2_{s,\zeta_s}(s_{t+1}, \tilde{a}_{t+1})) \\
\tilde{r}_{t+1} &\sim \mathcal{N}(\hat{f}_{r,\zeta_r}(s_{t+1}, \tilde{a}_{t+1}, \tilde{s}_{t+2}), \hat{\sigma}^2_{r,\zeta_r}(s_{t+1}, \tilde{a}_{t+1}, \tilde{s}_{t+2})) \\
\tilde{d}(\tilde{s}_{t+2}) &\sim \mathcal{N}(\hat{f}_{d,\zeta_d}(\tilde{s}_{t+2}), \hat{\sigma}^2_{d,\zeta_d}(\tilde{s}_{t+2}))
\end{aligned}
\tag{9}
$$

We apply the distributional expansion starting from $r_t, s_{t+1}$ to acquire the trajectory $\tilde{\tau}_{\zeta_r,\zeta_s,\zeta_d,\theta',H}(r_t, s_{t+1}) = (r_t, s_{t+1}, \tilde{a}_{t+1}, \tilde{r}_{t+1}, \tilde{s}_{t+2}, ..., \tilde{s}_{t+H+1}, \tilde{a}_{t+H+1})$, based on which we write DVE as equation. 10.

$$
\begin{aligned}
&\hat{Q}^{\text{DVE}}_{\zeta_r,\zeta_s,\zeta_d,\theta',\phi',H}(r_t, s_{t+1}) \\
&= r_t + \sum_{t'=t+1}^{t+H} \gamma^{t'-t} d_{t,t'} \tilde{r}_{t'} + \gamma^{H+1} d_{t,t+H+1} \hat{Q}_{\phi'}(\tilde{s}_{t+H+1}, \tilde{a}_{t+H+1})
\end{aligned}
\tag{10}
$$

$$
\text{with } d_{t,t'} = (1 - d(s_{t+1})) \prod_{k=t+2}^{t'} (1 - \tilde{d}(\tilde{s}_k))
$$

We then keep track of an ensemble of the combination of the parameters $\{\zeta_r, \zeta_s, \zeta_d, \phi'\}$. For each group of parameters we use an ensemble of $N$ parameters, which gives us $4N$ parameters in all. Then we select a random combination of four integers $\{i, j, k, l\}$, which gives the parameter combination of $\{\zeta_{r,i}, \zeta_{s,j}, \zeta_{d,k}, \phi'_l\}$. By switching the combination of integers we acquire an ensemble of DVE estimation. Then we count the average and the variance on the ensemble of DVE, and by subtracting a certain proportion($\alpha$) of the standard variance, we acquire a lower bound of DVE estimation. We call this estimation of value function the $\alpha$-confidence lower bound($\alpha$-CLB), written as equation. 11.

$$
\begin{aligned}
\hat{Q}^{\alpha-CLB}_H(r_t, s_{t+1}) &= \hat{\mathbb{E}}_{\zeta_r,\zeta_s,\zeta_d,\phi'} \left[ \hat{Q}^{\text{DVE}}_{\zeta_r,\zeta_s,\zeta_d,\theta',\phi',H}(r_t, s_{t+1}) \right] \\
&- \alpha \sqrt{\hat{\mathbb{V}}_{\zeta_r,\zeta_s,\zeta_d,\phi'} \left[ \hat{Q}^{\text{DVE}}_{\zeta_r,\zeta_s,\zeta_d,\theta',\phi',H}(r_t, s_{t+1}) \right]}
\end{aligned}
\tag{11}
$$

Subtraction of variances is commonly used in risk-sensitive RL(Sato & Kobayashi, 2000; Pan et al., 2019; Reddy et al., 2019). The motivation is straight forward, we try to suppress the utility of the

high-variance trajectories, in order to avoid possible risks. However, the $\alpha$ here is left undecided. We will come to this problem in the next part.

Finally, we define RAVE, which adopts the similar interpolation among different horizons as STEVE based on DVE and CLB, shown in equation. 13.

$$\hat{Q}_{target}(r_t, s_{t+1}) \leftarrow \hat{Q}^{\text{RAVE}}(r_t, s_{t+1}) = \frac{\sum_{H=0}^{H_{max}} \omega_H \hat{Q}_H^{\alpha-\text{CLB}}(r_t, s_{t+1})}{\sum_{H=0}^{H_{max}} \omega_H},$$

$$\text{with } \omega_H = \hat{\mathbb{V}}_{\zeta_r, \zeta_s, \zeta_d, \phi'} \left[ \hat{Q}_{\zeta_r, \zeta_s, \zeta_d, \theta', \phi', H}^{\text{DVE}}(r_t, s_{t+1}) \right]^{-1} \tag{12}$$

While adopting the lower confidence bound may introduce the bias of underestimation, it makes the policy less preferable to actions with large variance of future return.

## 5.2 Adaptive Confidence Bound

An unsolved problem in RAVE is to select proper $\alpha$. The requirement of risk aversion and exploration is somehow competing: risk aversion seek to minimize the variance, while exploration searches states with higher variance. To satisfy both requirements, previous work proposed two competing agents, and each will make decision for a short amount of time(Pan et al. (2019)). Here we propose another solution to this problem. We argue that the agent needs to aggressively explore at the beginning, and it should get more risk sensitive as the model converges. A key indicator of this is the epistemic uncertainty. The epistemic uncertainty measures how well our model get to know the state space. In MBRL and Hybrid-RL, there is a common technique to easily monitor the epistemic uncertainty, by evaluating the ability of the learned environment model to predict the consequence of its own actions(Pathak et al., 2017).

Following this motivation, we set the confidence bound factor to be related to its current state and action, denoted as $\alpha(s, a)$. We want $\alpha(s, a)$ to be larger when the environment model could perfectly predict the state to get more risk sensitive, and smaller when the prediction is noisy to allow more exploration. We have

$$\alpha(s_t, a_t) = max\{0, \alpha(1.0 - \frac{1}{Z} ||\mathbb{E}_{\zeta_s}[\hat{f}_{s, \zeta_s}(s_t, a_t)] - s_{t+1}||^2)\}, \tag{13}$$

where $Z$ is a scaling factor for the prediction error. With a little abuse of notations, we use $\alpha$ here to represent a constant hyperparameter, and $\alpha(s, a)$ is the factor that is actually used in $\alpha$-CLB. $\alpha(s_t, a_t)$ picks the value near zero at first, and gradually increases to $\alpha$ with the learning process.

## 6 Experiments and Analysis

We evaluate RAVE on continuous control environments using the MuJoCo physics simulator(Todorov et al., 2012). The baselines includes the model-free DDPG and STEVE that currently yields the SOTA Hybrid-RL performance in MuJoCo. We also align our performance with the SOTA MFRL methods including twin delayed deep deterministic (TD3) policy gradient algorithm(Fujimoto et al., 2018), soft actor-critic(SAC) algorithm(Haarnoja et al., 2018), and proximal policy optimization(PPO)(Schulman et al., 2017), using the implementation provided by the authors. To further demonstrate the robustness in complex environments, we also evaluate RAVE on OpenAI's Roboschool (Klimov & Schulman, 2017), where STEVE has shown a large improvement than the other baselines.We detail hyper-parameters and the implementation in the supplementary materials.

## 6.1 Experimental Results

We carried out experiments on eight environments shown in fig. 3. Among the compared methods, PPO is the only on-policy updating method, which has very poor sample-efficiency compared with either STEVE or off-policy MFRL methods, as PPO needs a large batch size to learn stably(Haarnoja et al., 2018). DDPG achieves quite high performance in HalfCheetah-v1 and Swimmer-v1, but fails on almost all the other environments, especially on challenging environments such as Humanoid.

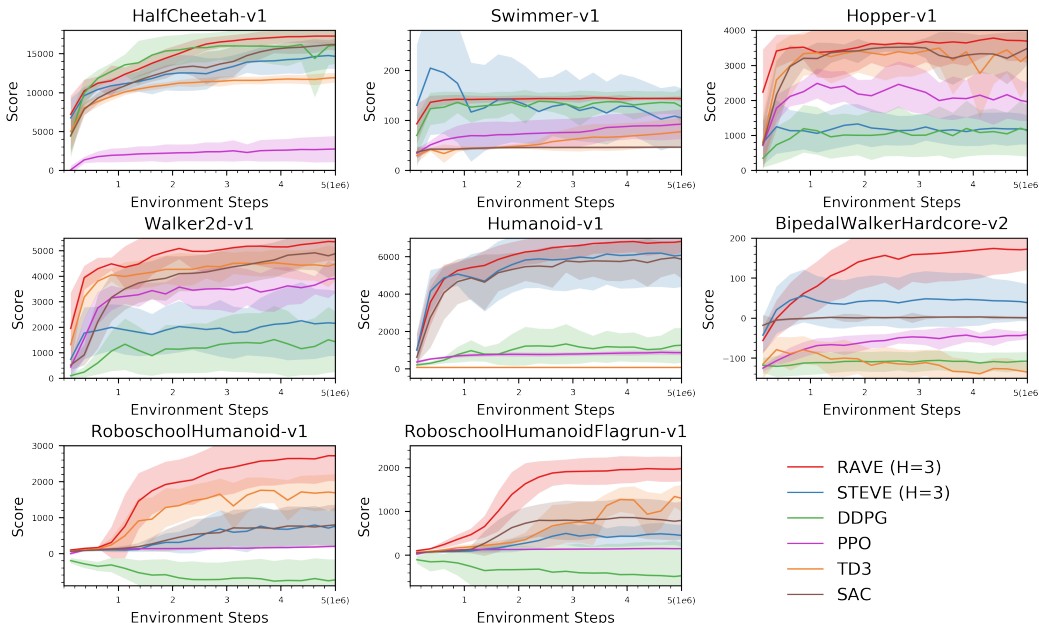

Figure 3: Average return over environment frames in MuJoCo and Roboschool environments. Each experiment is run four times.

In Hopper-v1 and Walker2d-v1, STEVE can not compare with TD3 and SAC, which yields quite good performance in many environments. However, RAVE performed favorably in most environments in both asymptotic performance and the rising speed(meaning sample efficiency), except for HalfCheetah-v1 and Swimmer-v1, where DDPG has already achieved satisfying performance and the margin between DDPG and Hybrid-RL is not that large.

## 6.2 ANALYSIS

**Distribution of Value Function Approximation**. To investigate whether the proposed method predicts value function more precisely, we plot the of the predicted values $\hat{Q}$ against the ground truth values of Hopper-v1 in fig. 4. The ground truth value here is calculated by directly adding the rewards of the left trajectory, thus it is more like a monte carlo sampling from ground truth distribution, which is quite noisy. To better demonstrate the distribution of points, we draw the confidence ellipses representing the density. The points are extracted from the model at environment steps of 1M. In DDPG and STEVE, the predicted value and ground truth aligned poorly with the ground truth, while RAVE yields better alignments, though a little bit of underestimation.

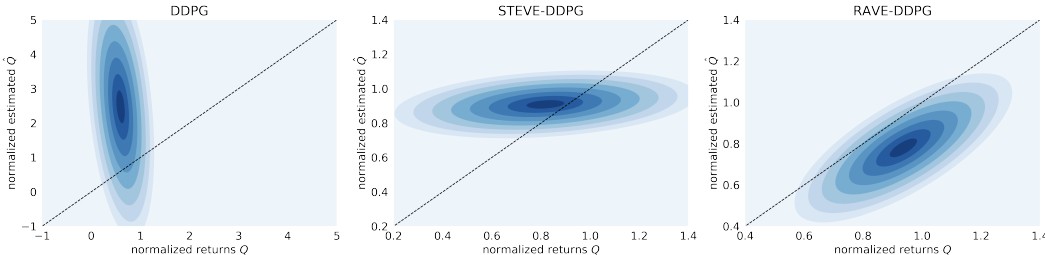

Figure 4: Confidence ellipses of the distribution of estimated values $\hat{Q}$ versus the ground truth in Hopper-v1. The points are extracted from environment steps of 1M, each includes statistics from 10,000 points. The x-axis and y-axis represent the statistical cumulative discounted returns(ground truth) and the predictive Q-values(both are normalized), respectively.

.

**Investigation on dynamic confidence bound**. In order to study the role played by the $\alpha$-confidence lower bound separately, we further carried out series of ablative experiments in Hopper-v1 environment. We compare RAVE($\alpha$ =constant), RAVE(dynamic $\alpha(s, a)$) and other baselines in chart. For all the algorithms, we set $H_{max} = 3$, and the experiments are replicated for 4 times.

From fig. 5(a) we can see that $\alpha = 0$(which means ensemble of DVE only) already surpasses the performance of STEVE in Hopper-v1, showing that modeling aleatoric uncertainty through PE indeed benefits the performance of value expansion. Larger margin is attained by introducing $\alpha$-CLB. A very large $\alpha$(such as constant $\alpha = 2.0$, which means lower CLB) can quickly stabilize the performance, but its performance stayed low due to lack of exploration, while a smaller $\alpha$(constant $\alpha = 0.0, 0.5$ generates larger fluctuation in performance. The dynamic adjustment of $\alpha$ facilitates quick rise and stable performance.

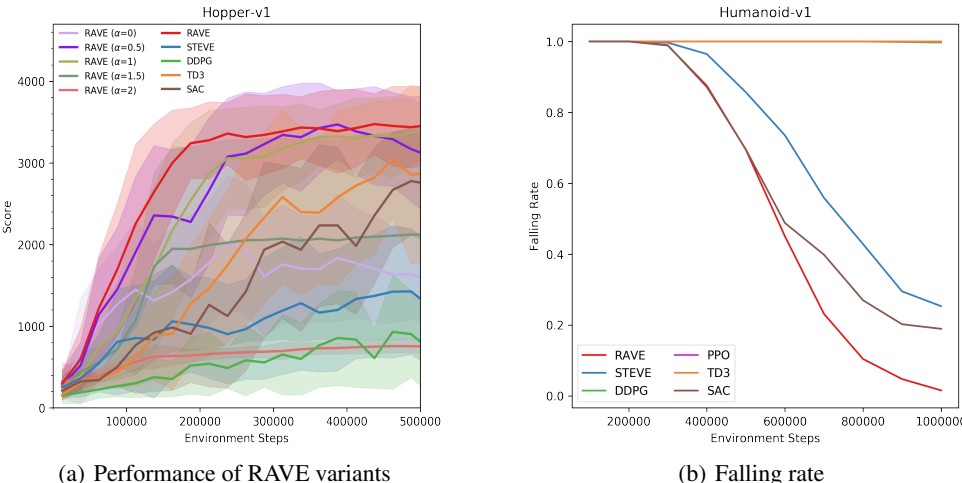

(a) Performance of RAVE variants         (b) Falling rate

Figure 5: Examining the performance of RAVE variants and robustness of the learned policy. Each experiment is run four times. (a) Performance of RAVE with different $\alpha$-CLB.(b) The falling rate of various algorithms over the first 1M environment steps. Each point is evaluated on the outcome of 1000 episodes.

**Analysis on Robustness**. We also investigate the robustness of RAVE and baselines on the most challenging Mujoco environment, Humanoid-v1. Humanoid-v1 involve complex humanoid dynamics, where the agent is prone to fall. We evaluate the robustness with the possibility of falling by the learned policy. As shown in fig.5(b), RAVE achieves the lowest falling rate compared with the baselines.

**Computational Complexity**. A main concern toward RAVE may be its computational complexity. On the one hand, hybrid-RL involves the usage of environment model which introduces additional computational cost. On the other hand, RAVE and STEVE involves ensemble of trajectory rollouts, which is a little bit costly. We keep the ensemble size the same as STEVE and the details about the hyper-parameter can be found in the supplements.

For the training stages, the additional training cost of RAVE compared with STEVE comes from modeling aleatoric uncertainty and additional sampling cost. We tested the training speed of STEVE and RAVE, and the time for RAVE to finish training 500 batches with a batch size 512 is 13.20s, an increase of 24.29%, compared to STEVE(10.62s). The time reported here is tested in 2 P40 Nvidia GPUs with 8 CPUs(2.00GHz). For the inference stages, RAVE charges exactly the same computational resources just as the other model-free actor critic methods as long as the model architecture of the policy is equal, which is a lot more cost efficient compared with MBRL that adopts a planning procedure.

Also we want to emphasize here that the computation complexity is typically less important compared with sample efficiency, as the interaction cost matters more than computational cost in training procedure.

# 7 CONCLUSION

In this paper, we raise the problem of incomplete modeling of uncertainty and insufficient robustness in model-based value expansion. We introduce ensemble of probabilistic models to approximate the environment, based on which we introduce the distributional value expansion(DVE), $\alpha$-Confidence Lower Bound, which further leads to RAVE. Our experiments demonstrate the superiority of RAVE in both sample efficiency and robustness, compared with state-of-the-art RL methods, including the model-free TD3 algorithm and the Hybrid-RL STEVE algorithm. We hope that this algorithm will facilitate the application of reinforcement learning in real-world, complex and risky scenarios.

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

# A  TRAINING AND IMPLEMENTATION DETAILS

## A.1  NEURAL NETWORK STRUCTURE

We used rectified linear units (ReLUs) between all hidden layers of all our implemented algorithms. Unless otherwise stated, all the output layers of model have no activation function.
**RL Models**. We implement model-based algorithms on top of DDPG, with a policy network and a Q-value network. The policy network is a stack of 4 fully-connected(FC) layers. The activation function of the output layer is *tanh* to constrain the output range of the network. The Q-value network takes the concatenation of the state $s_t$ and the action $a_t$ as input, followed by four FC layers.

**Dynamics Models**. We train three neural networks as the transition function, the reward function and the termination function. We build eight FC layers for the transition approximator, and four FC layers for the other approximators. The distributional models $\mathcal{N}(\hat{f}, \hat{\sigma}^2)$ in RAVE use the similar model structure except that there are two output layers corresponding to the mean and the variances respectively.

## A.2  PARALLEL TRAINING

We use distributed training to accelerate our algorithms and the baseline algorithms. Following the implementation of STEVE, we train a GPU learner with multiple actors deployed in a CPU cluster. The actors reload the parameters periodically from the learner, generate trajectories and send the trajectories to the learner. The learner stores them in the replay buffer, and updates the parameters with data randomly sampled from the buffer. For the network communication, we use Ray(Moritz et al., 2018) to transfer data and parameters between the actors and the learner. We have 8 actors generating data, and deploy the learner on the GPUs. DDPG uses a GPU, and model-based methods uses two: one for the training of the policy and another for the dynamics model.

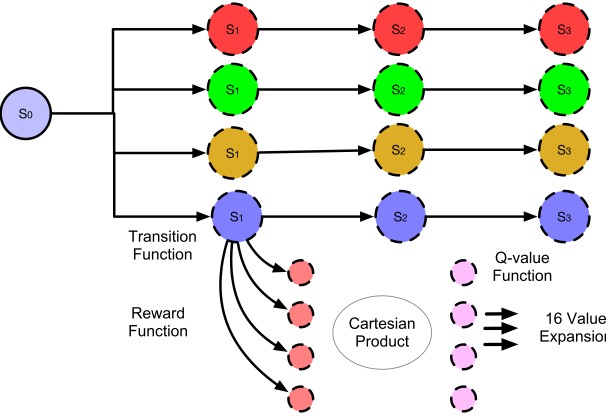

Figure 6: An illustration of the rollout result with $N = 4, H_{max} = 3, P = 4$. For each state of the trajectory, there are 16 candidate targets.

## A.3  ROLLOUT DETAILS

We employ the identical method of target candidates computation as STEVE, except we image a rollout with an ensemble of probabilistic models . At first, we bind the parameters of transition model($\zeta_{s,i}$) to the termination model($\zeta_{d,i}$). That is, we numerate the combination of three integers $\{i, j, k | i, j, k \in [1, N]\}$, which gives us an ensemble of $N^3$ parameters $\{\zeta_{r,j}, \zeta_{s,i}, \zeta_{d,i} \phi'_k\}$. The actual sampling process goes like this: For each $H \in [0, H_{max}]$, we first use the transition model($\zeta_{s,i}$) and the termination model($\zeta_{d,i}$) to image a state-action sequence $\{s_{t+1}, \tilde{a}_{t+1}, \tilde{s}_{t+2}, \tilde{a}_{t+2}, ..., \tilde{s}_{t+H+1}, \tilde{a}_{t+H+1}\}$; Based on the state-action sequence, we use reward function($\zeta_{r,j}$) to estimate the rewards ($\hat{r}_{t'} = \hat{f}_{r,\zeta_{r,j}}(\tilde{s}_{t'}, \tilde{a}_{t'}, \tilde{s}_{t'+1})$) and the value function($\phi'_k$) to predict the value of the last state($\hat{Q}_{\phi'_k}(\tilde{s}_{t+H+1}, \tilde{a}_{t+H+1})$)(fig. 6). In total we predict $N^3(H_{max} + 1)$ combination of rewards and value functions in both RAVE and STEVE.

## B    HYPER-PARAMETERS FOR TRAINING

We list all the hyper-parameters used in our experiments in table 1.

Table 1: Table of hyper-parameters

| Hyper-parameter | Value | Description |
|---|---|---|
| $B$ | 512 | Batch size for training the RL, and also the dynamics model |
| $N_{rpm}$ | 1e6 | Size of the replay buffer storing the transitions |
| $lr_\pi$ | 3e-4 | Learning rate of the training policy |
| $lr_Q$ | 3e-4 | Learning rate of the Q-value function |
| $lr_D$ | 3e-4 | Learning rate of the dynamics model |
| $\epsilon$ | 0.05 | Probability of adding a Gaussian noise to the action for exploration |
| $H_{max}$ | 3 | Maximum horizon length for value expansion |
| $N$ | 4 | Ensemble size of the value function and environment models |
| $F$ | 10000 | Number of collected frames to pretrain the dynamics model before training the policy |
| $Z$ | 1 | Scaling factor for the prediction error |
| $alpha$ | 1.5 | Confidence lower bound in equation.13 |

