# OpenReview forum: "Risk Averse Value Expansion for Sample Efficient and Robust Policy Learning"
_ICLR.cc/2020/Conference — Reject_

### Official Review · AnonReviewer4 · 2019-10-23
**Official Blind Review #4**

**Rating:** 3

**Review:**

This paper proposes a novel deep reinforcement learning algorithm at the intersection of model-based and model-free reinforcement learning: Risk Averse Value Expansion (RAVE). Overall, this work represents a significant but incremental step forwards for this "hybrid"-RL class of algorithms. However, the paper itself has significant weaknesses in its writing, analysis, and presentation of ideas.

The main strength of this work is its empirical section. The proposed algorithm is fairly compared against many relevant baselines on a variety of continuous control tasks, on both the Mujoco and Roboschool simulators, and demonstrates significant performance increases across all of them. The visualization in Figure 4 is interesting, and provides good insight into the issues with DDPG and STEVE, and the reasons for the success of RAME. Based on these results, the authors have convinced that RAME is a state-of-the-art algorithm.

However, in spite of its strong performance on benchmarks, I believe that this paper needs a significant overhaul/rewrite before publication.

One major criticism I have is on the authors' treatment of the different types of uncertainty. The introductory sections spend a large amount of time on the differences between aleatoric and epistemic uncertainty, and various other related concepts. But when it comes to the actual new algorithm, that the authors only barely touched upon the (very important) point that V[Q^DVE] is actually a *mix* of epistemic and aleatoric uncertainties. When minimizing this term, it's not clear whether epistemic or aleatoric uncertainty is actually being reduced. (Based on the experiments, which report only expected values, not CVaR or anything of the sort, it seems the authors care only about reducing epistemic uncertainty; if so, it's unclear why they choose to minimize a term which includes aleatoric uncertainty too.) A more principled understanding of this quantity seems essential to this line of work. Similarly, "risk" typically refers to aleatoric uncertainty, but the risk-senstive component of RAME computes the confidence lower bound w.r.t. a mix of aleatoric and epistemic uncertainty. Calling this algorithm "risk sensitive" is likely to generate confusion, in my opinion.

A few other points on presentation of ideas:
- Switching from a deterministic to a stochastic model is a trivial extension of MVE/STEVE, and way too much time is spent on this point (even going so far as to name a new algorithm!). Section 4 contains no new insight, beyond "deterministic models can be bad in stochastic environments", which is obvious. Consider re-thinking your experiments for this section to help readers better understand *what* goes wrong.
- In my opinion, RAME is as much a successor to TD3 as it is to STEVE. Taking the lower outcome of an ensemble of size 2 is equivalent to applying a penalty based on the stdev of the outcome distribution, with α=1. The introduction of the paper should be re-written to clearly point out that that RAME ~= TD3 + STEVE.
- I'd also like to see a lot more ablations, on at least a few environments. There are at least four factors that need to be teased apart: 1) deterministic vs stochastic model 2) lower confidence bound penalty 3) adaptive α for LCBP 4) STEVE reweighting. Does RAME require all of these elements to perform well? How do these elements interact?

Some feedback on the writing:
- There are various small factual errors. For example, in the very first paragraph, the Dyna algorithm is attributed to Kuturach et al, instead of Sutton (1990). (The algorithm from Kuturach is ME-TRPO.)
- The notation is not very good; there are lots of symbols flying around everywhere, and it makes the ideas (which are fundamentally very simple) a bit difficult to parse. For example, naming each parameter-set individually, everywhere, is unnecessary.
- There are some strange non-sequiturs and overall lack of flow.

I think that this work has a lot of potential, and am especially impressed by the empirical results. I recommend rejection in its current form, but hope to see a revised version of this work appear at a conference in the near future.

**Experience Assessment:**

I have published one or two papers in this area.

**Review Assessment: Checking Correctness Of Derivations And Theory:**

I assessed the sensibility of the derivations and theory.

**Review Assessment: Checking Correctness Of Experiments:**

I assessed the sensibility of the experiments.

**Review Assessment: Thoroughness In Paper Reading:**

I read the paper thoroughly.

---

### Official Review · AnonReviewer1 · 2019-10-24
**Official Blind Review #1**

**Rating:** 3

**Review:**

In this work , the authors combine model-based value expansion(MVE) with
model-free reinforcement learning and also take into account the high-order stochastic characteristics
of the environment to make the value expansion algorithm in modeling error propagation of dynamics risk-averse.
They propose a novel Hybrid-RL method, namely the Risk Averse Value Expansion(RAVE), that uses an ensemble of probabilistic dynamics models to generate imaginative rollouts and to model risk aversion
of risks by estimating the lower confidence bound of the ensemble. They showed that by taking the risk (in terms of variance) of the dynamic propagation error into the account, RAVE can achieve comparable performance with other state-of-the-art baselines in benchmark experiments including MuJoCo and robo-school. Also they showed that RAVE reduced the variance of the return and thus prevented catastrophic consequences such as falling.

I found this work interesting as the authors try to take the uncertainty of probabilistic dynamics model into account for estimating the value function and its confidence bounds in model-free RL. Utilizing such confidence interval of values, they can have a way of doing exploration while being risk-averse. While similar approaches of modeling the mean and variance of Q functions can be found in some existing work, such as boot-strapped DQN (Osband'16), and efficient exploration via Bayesian DQN (Azizzadenesheli'18), none of these work model the variance of the Q-function using the error propagation of the dynamics model. Through extensive experiments, it has also demonstrated the effectiveness of RAVE in terms of achieving good performance, approximation error in value estimation, as well as robustness to failure in standard mujoco benchmarks, which provides readers some detailed understanding on how this risk-averse uncertainty modeling/propagation technique helps in exploration. However, while i found this idea interesting, it appears to me that the current work is still quite preliminary (without theoretical concentration inequality based variance bounds for guiding exploration), and empirically, it would be great to compare this method with the aforementioned bootstrapping/bayesian based approaches. My other major comment is on claiming the proposed method to be "risk-averse", because in RL, risk-averse methods are commonly known to not only optimize the expected return but also to provide guarantees to other moments of the return, such as variance of CVaR. However, while the method used variance of the expected value (due to the error propagation of the dynamics) for exploration, I am not seeing the risk-averse optimization criteria being studied here. Therefore calling the RL method risk-averse might be a little mis-leading.

**Experience Assessment:**

I have read many papers in this area.

**Review Assessment: Checking Correctness Of Derivations And Theory:**

I assessed the sensibility of the derivations and theory.

**Review Assessment: Checking Correctness Of Experiments:**

I assessed the sensibility of the experiments.

**Review Assessment: Thoroughness In Paper Reading:**

I made a quick assessment of this paper.

---

### Official Review · AnonReviewer2 · 2019-10-26
**Official Blind Review #2**

**Rating:** 6

**Review:**

This paper considers the problem of high function approximation errors facing the stochastic environments when trying to combining model-based reinforcement learning (RL) with model-free RL. The paper begins by showing that previous methods like model-based value expansion (MVE) and stochastic ensemble value expansion (STEVE) can perform even worse than the pure model-free DDPG algorithm in a rather noisy environment. Following the ideas of MVE and STEVE, It then proposes a risk averse value expansion (RAVE) to replace the target Q function in the actor-critic algorithm, which is built upon an ensemble of probabilistic models (PE) and adopt the lower confidence bound as a surrogate of the target value as in the risk-sensitive RL. A simple yet intuitive approach for adaptively selecting the confidence bound \alpha is also proposed. The experiments show that RAVE does improve over the state-of-the-art algorithms in several different environments, with a better draw-down control. In general, this paper is well-written and the idea of RAVE is novel as far as I know. But since I'm not very familiar with the specific literature of combing model-based and model-free RL, and since the idea of RAVE is relatively straightforward (but admittedly practically powerful and theoretically interesting), I choose to give a conservative accept to account the possibility that some existing works have followed very similar approaches.

Some minor comments: 1) What is the loss of the negative log-likelihood in (4) for? 2) The authors may want to explain clearly what is the variance indicating in (8) more clearly, although one can guess that it is closely related to (5).

**Experience Assessment:**

I do not know much about this area.

**Review Assessment: Checking Correctness Of Derivations And Theory:**

I assessed the sensibility of the derivations and theory.

**Review Assessment: Checking Correctness Of Experiments:**

I assessed the sensibility of the experiments.

**Review Assessment: Thoroughness In Paper Reading:**

I read the paper at least twice and used my best judgement in assessing the paper.

---

> ### Author Response · Authors · 2019-11-14
> **Reply**
>
> We would like to thank the reviewer for the constructive comments. We are glad that the reviewer found the proposed method useful and novel.
>
> Regarding the comments:
>
> > What is the loss of the negative log-likelihood in (4) for?
> RAVE uses a probabilistic model to predict the reward, and we can maximize log-likelihood or minimize the negative log-likelihood to maximize its probability over the reward samples.
>
>
> >The authors may want to explain clearly what is the variance indicating in (8), although one can guess that it is closely related to (5).
>
>
> Thanks for the suggestion. We would rewrite the symbols in(8) to make it more clear.

---

> > ### Comment · AnonReviewer2 · 2019-11-14
> > **Thanks for the clarification, but make sure to include them in the updated draft**
> >
> > Thanks for the explanations. For the loss of the negative log-likelihood, yes I understand it's used for inferring the parameters from the reward samples. However, you should make it clearer how this idea is used and applied in RAVE more explicitly. In particular, since you mentioned that RAVE uses the PE model in Section 5, you should clearly state how you infer/estimate those parameters in the PE model. However, the log-likelihood is only mentioned for the non-ensemble case, which does not seem to be used later in RAVE. Btw, a quick suggestions is to add a sentence to clarify which model is used throughout the paper for RAVE right after Section 3.3, to reduce confusion.
> >
> > To make the paper more accessible, I have the following additional suggestions: 1) add an algorithm framework (maybe in the appendix) summarizing the entire framework of RAVE, including how you infer the parameters, etc., i.e., essentially the entire approach you apply in the experiments; 2) change the rather non-standard and weird notation of putting parameter below the function (like in equation 4) to subscripts, or putting those parameters to the right hand side of the function and include them in the brackets as used in the standard function notation; 3) make sure that the brackets before the equation reference is separated from the previous words by a blank.

---

### Official Review · AnonReviewer3 · 2019-11-02
**Official Blind Review #3**

**Rating:** 3

**Review:**

Summary
This paper expands on previous work on hybrid model-based and model-free reinforcement learning. Specifically, it expands on the ideas in Model-based Value Expansion (MVE) and Stochastic Ensemble Value Expansion (STEVE) with a dynamically-scaled variance bias term to increase risk aversion over the course of learning, which the authors call Risk Averse Value Expansion (RAVE). Experimental results indicate notable improvements over their selected model-free and hybrid RL baselines on continuous control tasks in terms of initial learning efficiency (how many environment steps are needed to achieve a particular level of performance), asymptotic performance (how high the performance is given the same large number of environment steps), and avoidance of negative outcomes (how infrequently major negative outcomes are encountered over the course of training).

Review
The core contribution of the paper is an extension to STEVE that uses an idea from risk-averse RL of biasing the underlying estimators away from high-variance predictions, and adds a dynamic weight to that bias.

Strengths
- The addition of a risk-aversion term to STEVE is a good contribution to the literature on safe RL. While it may be possible to criticize this contribution as somewhat trivial, I am disinclined to do so, as finding simple ideas that are effective is the hallmark of an important contribution, in my opinion.
- Under the assumption that the baselines are fair, the empirical results show substantial improvements for a good selection of challenging tasks along three metrics: initial efficiency, asymptotic performance, and avoidance of negative outcomes.
- The paper provides a careful and detailed section on the relevant preliminaries, giving precise notation that is expanded step-by-step from basic actor-critic approaches all the way through the description of RAVE. (Although see my comment below about possibly making the notation more concise.)
- Overall, the paper’s presentation is clear and natural. (Although see my comment below about a strong need for proofreading.)

Weaknesses
- The paper opens with a claim that MBRL has worse performance than MFRL in environments with “noisy environments and long trajectories”. However, recent work [1] has shown that MBRL can outperform MFRL on many of the tasks considered in this paper using far fewer environment steps, even when training directly from pixels.
- The experiments are all based on observations of the true state vectors rather than pixels, which dramatically simplifies working with imagination rollouts, as the dimensionality is so much lower. This hides the high computational cost and modeling challenges involved in using imagination-based MBRL or hybrid RL approaches in real-world environments, and (in my opinion) is a major shortcoming of this line of research (not only of this paper). Since the DDPG family of algorithms doesn’t have to do rollouts, they have a strong advantage over this type of approach on more realistic settings where the true state is unknown and must be inferred from high-dimensional observations. If experiments are not going to be done using pixels, the discussion should at least mention the trade-offs involved between the proposed algorithm and the baselines in that setting.
- The core baseline of the paper is DDPG, which is unnecessarily weak. D4PG [2] came out a year and a half ago and has the same high-level properties as DDPG, but outperforms it on all of the Mujoco tasks considered in this paper. Additionally, it includes some of the ideas presented in this work, but in a model-free setting, including a distributional treatment and multi-step rollouts, so it is easy to imagine that some of the experimental gains presented here would be erased when using it as a baseline.
- The use of ensembling is another confounder in the experiments. Ensembling models almost always yields an improvement, so any technique that relies on some form of ensembling needs to additionally demonstrate that the gains presented are not solely due to ensembling. In this case, the experiments with STEVE and RAVE should minimally be performed with different numbers of models in their ensembles.
- (Minor) Similarly, the prediction horizon can have a large impact, and only STEVE and RAVE consider prediction horizons other than 1. Showing how the horizon affects performance would help make the comparisons more fair.
- Figure 1 shows a bias problem in some current approaches in a clear toy problem, but does not show whether RAVE addresses that problem. Even though this section precedes the presentation of RAVE, it is necessary to show that the proposed solution actually helps on the toy problem.
- There are a number of problems with the proposal for adaptively computing the alpha parameter. In general, such adaptive hyperparameters add a great deal of complexity to hyperparameter tuning, so such suggestions should be either strongly motivated by theory or by empirical results. Neither seems to be the case here.
  - Only one proposal for adaptive alpha (equation 13) is considered. Its justification is plausible, but it would be more convincing if other dynamic approaches were considered in the experiments. For example:
    - alpha(env_step) = alpha * env_step / max_env_step
    - alpha(s_t,a_t) = alpha * min{1.0, 1 / (Z * ||E[fhat(s_t,a_t)] - s_t+1||^2)}
    - The opposite of the proposed approach, where alpha starts high and gets lower over the course of training.
  - Indeed, a potentially useful quantity during evaluation, when s_t+1 is unknown, would be based on the variance of the predicted next state, rather than the difference of the expectation of the predicted next state and the true next state. E.g.:
    - alpha(s_t,a_t) = alpha / Var[fhat(s_t,a_t)]
    This formulation says that the precision of the prediction determines the confidence of the model, which is also intuitively reasonable (to me, anyway) and doesn’t rely on knowing the future.
  - (Minor) The adaptive variant appears to be only slightly better than alpha=0.5, but requires two hyperparameters that are unspecified in the main body of the paper — alpha and Z. The appendix lists those parameters, but no discussion is made on how much tuning was done to determine that pair of parameters.
  - (Minor) A plot of how the dynamic value of alpha changes during training would be useful, at least in the appendix.

Recommendation
In light of my comments above, I cannot currently recommend the acceptance of this paper at ICLR. However, I think that the core idea is likely to hold up under more careful experimental comparisons. If the authors submit a revised draft that addresses the substance of my concerns, I would be very likely to increase my rating. In particular, I would like to see much more careful experimental treatment of the idea, so that readers can have high confidence about the circumstances where RAVE is likely to be a good choice.

[1] Hafner et al., “Learning Latent Dynics for Planning from Pixels”. ICML 2019. https://arxiv.org/abs/1811.04551
[2] Barth-Maron et al., “Distributed Distributional Deterministic Policy Gradients”. ICLR 2018. https://arxiv.org/abs/1804.08617


Other Comments and Questions
- This paper needs a lot of proofreading. A few examples of errors that should be fixed before publication:
  - “equation. 2”: This should read “Equation 2”, “equation 2”, “Eq. 2” or “eq. 2”. The period in the last two options indicates that letters have been elided. The mistake of using a period where no elision has happened occurs throughout the paper.
  - “prophesy”: This is not the correct word. Just say “prediction”.
  - “image”: This is often used when the correct word would be either “imagine” or “imagination”.
  There are many other errors that could be fixed easily with the help of a native English speaker.
- The mathematical notation in sections 3 through 5 is precise, but it is also a bit heavy. Consider whether there would be any ambiguity added if, for example, {\hat Q}^DVE_{\zeta_s,\zeta_r,\zeta_d,\theta’,\phi’} were instead notated {\hat Q}^DVE_{\zeta,\theta’,\phi’}.
- Figure 2 adds nothing of value to the paper and should be removed.
- Consider comparing on DMControl, which is the same set of tasks as Mujoco, but the scores are standardized such that each task has a maximum reward of 1000 per episode.
- Why does DDPG get worse over time on the RoboSchool tasks? Without a clear explanation, it looks like a bug, and a bug like that calls into question the rest of the DDPG results as well.


**Experience Assessment:**

I have published one or two papers in this area.

**Review Assessment: Checking Correctness Of Derivations And Theory:**

I carefully checked the derivations and theory.

**Review Assessment: Checking Correctness Of Experiments:**

I carefully checked the experiments.

**Review Assessment: Thoroughness In Paper Reading:**

I read the paper thoroughly.

---

> ### Author Response · Authors · 2019-11-14
>
> We would like to thank the reviewer for the detailed review and constructive comments.
> We do agree that the paper needs proofreading, and we will address the error mentioned in the future version. We also agree that our method based on MVE and STEVE is a bit complex. However, we argue that the extension on this line is not trivial, because model-based value expansion provides an alternative way to incorporate the dynamics model into MBRL methods, instead of using the dynamics model for the generation of additional training data.
>
> Regarding the comments:
> > MBRL  and MFRL performance in "noisy environments and long trajectories."
> For model-based RL methods, the authors of [1] have analyzed theoretically the value expansion error brought by the increasing rollout length H. Our empirical results on the toy environment also demonstrates the weakness of MBRL methods in a noisy environment. However, the experimental results of the paper[2] show that MBRL can outperform MFRL. On explanation is that in MFRL, the performance gap between learning from pixels and density vectors is still huge. For example, when learning from density observations, the performance on Halfcheetah-v1 can reach the maximum score, while learning from pixels reaches less than 800 scores. Compared with MFRL methods, which have a lower score, the MBRL method has better performance when learning from pixels.
>
> > Learning from pixels.
> We agree that it might be hard to apply some MBRL methods in environments with pixel observations. However, we believe that these algorithms can be helpful in some real-world problems, where most of the sensor data is float vectors. (e.g., heat-sensor, sonar sensor). To address the issue of learning from pixels, predicting future rewards/values without predicting raw observations might be the right choice, like VPN[3] and I2A[4]. We agree that there is a trade-off between the proposed method and MFRL methods, and will discuss this issue in the future version.
>
> > DDPG is a weak baseline.
> We agree that DDPG is a weak baseline, and that is why we further use TD3 and SAC, the best MFRL methods, as the baseline methods. The main concern of not including D4PG is that the improvement of our method might come from a multi-step rollout or using an ensemble of models. We use the same hyper-parameters, such as rollout steps and the number of ensemble models for RAVE and STEVE in our experiments. The performance gap between STEVE and RAVE shows that the performance gain of RAVE does not rise from these two techniques.
>
> > The use of ensembling is another confounder.
> As we mentioned in the last paragraph, we use the same number of ensemble models for STEVE and RAVE, so it is clear the RAVE's gain does not rely on the ensemble technique, compared with STEVE. However, we agree that the impact of ensemble size on these methods is an interesting problem. We will study its importance in the future version.
>
> > The impact of the prediction horizon.
> The reason why only STEVE and RAVE consider prediction horizons other than 1 is that they are MBRL methods, while other baselines are MFRL methods.
>
> > If the proposed solution actually helps on the toy problem.
> Thanks for the suggestion. We would add the performance of rave on the toy environment in the future version.
>
> > Only one proposal for adaptive alpha is considered.
> Thanks for the suggestion. We would add experiments of various kinds of adjustments on alpha in the future version.
>
> > No discussion is made on how much tuning was done to determine alpha and Z.
> We set the maximum alpha to Z, and we maintain a variable recoding the recent range of the approximation error and normalize all the forward propagation error into (0, 1.5) in our experiment.
>
> >How the dynamic value of alpha changes during training.
> Thanks for the suggestion. We would add a figure plotting the change of alpha in the future version.
>
> > Consider comparing on DMControl.
> Thanks for the suggestion. We would consider the benchmarks in our future work.
>
> > Why does DDPG get worse over time on the RoboSchool tasks?
>
> We use the DDPG implementation provided by the authors of STEVE. On explanation of the worse performance of DDPG in these environments is that these tasks are more challenging than previous tasks. Roboschool also provided simple environments such as Halfcheetah, but we only compare the algorithms on the most challenging environments of this library.
>
> [1] Feinberg et al., "Model-Based Value Expansion for Efficient Model-Free Reinforcement Learning." ICML 2018. https://arxiv.org/pdf/1803.00101.pdf
> [2] Hafner et al., "Learning Latent Dynamics for Planning from Pixels." ICML 2019. https://arxiv.org/abs/1811.04551
> [3] Feinberg et al., "Imagination-Augmented Agents for Deep Reinforcement Learning." NIPS2017. https://arxiv.org/abs/1707.03497
> [4] Weber et al., "Imagination-Augmented Agents for Deep Reinforcement Learning." NIPS2017. https://arxiv.org/abs/1707.06203

---

> > ### Comment · AnonReviewer3 · 2019-11-14
> > **Looking forward to the new version**
> >
> > Thank you for the comments and clarifications! I look forward to seeing the new version.

---

### Decision · Program_Chairs · 2019-12-19

**Decision:**

Reject

**Comment:**

The authors propose to extend model-based/model-free hybrid methods (e.g., MVE, STEVE) to stochastic environments. They use an ensemble of probabilistic models to model the environment and use a lower confidence bound of the estimate to avoid risk. They found that their proposed method yields state-of-the-art performance over previous methods.

The valid concerns by Reviewers 1 & 4 were not addressed by the authors and although the authors responded to Reviewer 3, they did not revise the paper to address their concerns. The ideas and results in this paper are interesting, but without addressing the valid concerns raised by reviewers, I cannot recommend acceptance.